# The Plasmatic Aldosterone and C-Reactive Protein Levels, and the Severity of Covid-19: The Dyhor-19 Study

**DOI:** 10.3390/jcm9072315

**Published:** 2020-07-21

**Authors:** Orianne Villard, David Morquin, Nicolas Molinari, Isabelle Raingeard, Nicolas Nagot, Jean-Paul Cristol, Boris Jung, Camille Roubille, Vincent Foulongne, Pierre Fesler, Sylvain Lamure, Patrice Taourel, Amadou Konate, Alexandre Thibault Jacques Maria, Alain Makinson, Ivan Bertchansky, Romaric Larcher, Kada Klouche, Vincent Le Moing, Eric Renard, Philippe Guilpain

**Affiliations:** 1Montpellier School of Medicine, University of Montpellier, 34000 Montpellier, France; orianne.villard@inserm.fr (O.V.); nicolas.molinari@inserm.fr (N.M.); n-nagot@chu-montpellier.fr (N.N.); jp-cristol@chu-montpellier.fr (J.-P.C.); b-jung@chu-montpellier.fr (B.J.); c-roubille@chu-montpellier.fr (C.R.); p-fesler@chu-montpellier.fr (P.F.); s-lamure@chu-montpellier.fr (S.L.); p-taourel@chu-montpellier.fr (P.T.); a-maria@chu-montpellier.fr (A.T.J.M.); a-makinson@chu-montpellier.fr (A.M.); r-larcher@chu-montpellier.fr (R.L.); k-klouche@chu-montpellier.fr (K.K.); v-le_moing@chu-montpellier.fr (V.L.M.); e-renard@chu-montpellier.fr (E.R.); 2Department of Endocrinology, Diabetes, Nutrition, and INSERM 1411 Clinical Investigation Centre, Montpellier University Hospital, INSERM, 34000 Montpellier, France; i-raingeard@chu-montpellier.fr; 3Institute of Functional Genomics, CNRS, INSERM, University of Montpellier, 34000 Montpellier, France; 4Department of Infectious and Tropical Diseases, Montpellier University Hospital, 34000 Montpellier, France; d-morquin@chu-montpellier.fr; 5IMAG, CNRS, University of Montpellier, Montpellier University Hospital, 34000 Montpellier, France; 6Laboratory of Biochemistry, Montpellier University Hospital, 34000 Montpellier, France; 7Department of Intensive Care Medicine, Montpellier University Hospital, 34000 Montpellier, France; 8PhyMedExp, Université de Montpellier, INSERM, CNRS, 34000 Montpellier, France; 9Laboratory of Virology, Montpellier University Hospital, 34000 Montpellier, France; v-foulongne@chu-montpellier.fr; 10Department of Radiology, Montpellier University Hospital, 34000 Montpellier, France; 11Department of Internal Medicine—Multi-Organ Diseases, Local Referral Center for Auto-Immune Diseases, Montpellier University Hospital, 34000 Montpellier, France; a-konate@chu-montpellier.fr (A.K.); i-bertchansky@chu-montpellier.fr (I.B.); 12Department of Internal Medicine—‘DIAGORA Unit’, Montpellier University Hospital, 34000 Montpellier, France; 13IRMB, INSERM U1183, Montpellier University Hospital, 34000 Montpellier, France

**Keywords:** Covid-19, renin angiotensin aldosterone system, inflammation, endocrine, severity

## Abstract

Background. The new coronavirus SARS-CoV-2, responsible for the Covid-19 pandemic, uses the angiotensin converting enzyme type 2 (ACE2), a physiological inhibitor of the renin angiotensin aldosterone system (RAAS), as a cellular receptor to infect cells. Since the RAAS can induce and modulate pro-inflammatory responses, it could play a key role in the pathophysiology of Covid-19. Thus, we aimed to determine the levels of plasma renin and aldosterone as indicators of RAAS activation in a series of consecutively admitted patients for Covid-19 in our clinic. Methods. Plasma renin and aldosterone levels were measured, among the miscellaneous investigations needed for Covid-19 management, early after admission in our clinic. Disease severity was assessed using a seven-category ordinal scale. Primary outcome of interest was the severity of patients’ clinical courses. Results. Forty-four patients were included. At inclusion, 12 patients had mild clinical status, 25 moderate clinical status and 7 severe clinical status. In univariate analyses, aldosterone and C-reactive protein (CRP) levels at inclusion were significantly higher in patients with severe clinical course as compared to those with mild or moderate course (*p* < 0.01 and *p* = 0.03, respectively). In multivariate analyses, only aldosterone and CRP levels remained positively associated with severity. We also observed a positive significant correlation between aldosterone and CRP levels among patients with an aldosterone level greater than 102.5 pmol/L. Conclusions. Both plasmatic aldosterone and CRP levels at inclusion are associated with the clinical course of Covid-19. Our findings may open new perspectives in the understanding of the possible role of RAAS for Covid-19 outcome.

## 1. Introduction

A new coronavirus called SARS-CoV-2 is responsible for the pandemic of Covid-19, which has led to tens of thousands of deaths around the world so far [1]. Briefly, the disease develops in two phases: the first one is linked to the viral invasion, and the second one consists of a severe acute inflammatory immune response, including a “cytokine storm”, which results in severe morbidity and mortality, mainly related to lung injury [2]. In this context, the intensity of the inflammatory process contributes to the disease severity and the plasmatic level of C-reactive protein (CRP) (a biomarker of systemic inflammation) could represent a marker of poor outcomes in Covid-19 patients [3,4,5,6].

As observed with the SARS-CoV responsible for SARS 2003 [7], SARS-CoV-2 uses the angiotensin converting enzyme type 2 (ACE2) as a cellular receptor to infect cells. ACE2 is a physiological inhibitor of the renin angiotensin aldosterone system (RAAS) through the catabolism of angiotensin type 2 (Ang2) into angiotensin (1–7) peptide [8]. Ang2 can induce pro-inflammatory responses through its receptor AT1R, while ACE2 reduces anti-inflammatory reactions through its receptor MasR. The assessment of RAAS involvement in the course of Covid-19 in humans is not easy, due to the poor value of Ang2 and angiotensin (1–7) peptide assays in peripheral blood to investigate RAAS and ACE2 in affected patients. Therefore, we explored the levels of plasma renin and aldosterone as indicators of RAAS activation in a series of consecutively admitted patients for Covid-19 in our clinic. Relationships with the severity of disease course were investigated to assess whether RAAS activation could be considered as a biomarker of Covid-19 outcomes.

## 2. Patients and Methods

### 2.1. Study Design

In a series of consecutive patients with Covid-19 diagnosis, hormonal assays including plasma renin and aldosterone levels were performed among the miscellaneous investigations needed for the management of Covid-19, early after admission in our clinic. This study is called Dyhor-19 (dysfunctional hormone regulation during Covid-19) and its protocol was reviewed and approved by the University Hospital of Montpellier Institutional Review Board (IRB-MTP_2020_04_202000441, ClinicalTrials.gov identifier: application in process).

### 2.2. Outcome Measures

The disease severity was assessed using a seven-category ordinal scale (OS) [9], as follows: 1—not hospitalized, no limitation on activities; 2—not hospitalized, limitations on activities; 3—hospitalized, not requiring supplemental oxygen; 4—hospitalized, requiring supplemental oxygen; 5—hospitalized, on non-invasive ventilation or high flow oxygen devices; 6—hospitalized, on invasive mechanical ventilation or extracorporeal membrane oxygenation (ECMO) and 7—death.

The primary outcome of interest was the severity of patients’ clinical courses during hospitalization, defined as severe for an ordinal scale higher than 4 and corresponding to the transfer to the intensive care unit and death from all-causes. Clinical status using the seven-category ordinal scale was assessed at three different time points: (i) early after admission corresponding to the day of the biological test (Day 0), (ii) two days later (Day 2) and (iii) considering the maximum ordinal scale during the overall period of hospitalization (OS max).

### 2.3. Data Collection

The past medical history, clinical manifestations, comorbidities, treatment strategies, radiologic assessments and laboratory testing on admission were extracted from the electronic medical records. Disease severity was assessed by the OS, as indicated above. Diagnosis of Covid-19 was considered as suspected in patients with typical lung CT-scan lesions and negative SARS-CoV-2 PCR. 

Laboratory variables were tested with conventional methods, including routine blood tests: blood count, renal function, inflammatory markers. Determination of CRP was performed using a Cobas8000/e502^®^ analyzer (Roche Diagnostic, Meylan, France) using the immunoturbidimetric method with reagents from Roche (total CV imprecision results in the laboratory = 3%). Renin and aldosterone were determined on an IDS-iSYS multi-discipline automated system (ImmunoDiagnosticSystem, Boldon, United Kingdom) using kits from IDS (total CV imprecision results in the laboratory 5% and 6% for renin and aldosterone respectively). A cutoff of 102.5 pmol/L was the lower limit of plasma aldosterone detection in our condition. Plasma cortisol and adrenocorticotropic hormone (ACTH) levels were measured by automated electrochemiluminescence assays (Cobas 8000, Roche, Basel, Switzerland). Laboratory confirmation of SARS-CoV-2 infection was determined by reverse transcription-PCR from nasopharyngeal swab specimens. 

### 2.4. Statistical Analyses

Categorical variables were described as frequency rates and percentages, analyzed using the Chi-squared test or Fisher’s exact test. Continuous variables were described using mean and standard deviation (SD). Means for continuous variables were compared using Student t-test or Mann–Whitney test according to the data distribution. Due to skewed distribution, biological variables were presented with median (min–max), and median difference (Hodges–Lehamann estimator). A logistic regression was used for the analysis of the main criteria with odds ratio of disease severity adjusted on the delay from admission. Covariates were selected in a backward selection procedure if *p* < 0.15 in the univariate analysis and then presented as adjusted odds ratios (ORs). Potential confounding factors were investigated by testing differences between groups. Studying the relationship between variables was done using Spearman correlation. Statistical analyses were performed using SAS Enterprise Guide, version 7.3 (SAS Institute, Cary, NC, USA) and GraphPad Prism, version 8.4.2 for Mac Os (GraphPad Software, San Diego, CA, USA). 

## 3. Results

### 3.1. Patients and Clinical Course

Forty-four patients were included in the study during the period from 26 March to 20 April 2020. Clinical characteristics of the patients are described in Table 1 and Appendix A. The median age of patients was 66.5 years (interquartile range (IQR, 53 to 75.3 years), and 38.6% of patients were women. The median interval time between symptom onset and baseline laboratory tests was 8 days (IQR, 5 to 10 days). At inclusion, 12 patients had mild clinical status (OS ≤ 3), 25 moderate clinical status (OS = 4) and 7 severe clinical status (OS ≥ 5). On Day 2, 17 patients met a mild clinical status including 2 patients who returned home whereas severe clinical status concerned 12 patients including 2 deaths. During the overall period of hospitalization, estimation of OS max showed that 15 patients (34.1%) met a severe clinical course including 6 deaths, and 6 patients requiring invasive mechanical ventilation. Of the 29 patients with mild or moderate clinical status, 19 required oxygen support. Clinical courses according to the ordinal scale are described in Table 2.

At inclusion, on Day 0, patients with severe clinical course (OS max ≥ 5) had more frequently a thyroid or chronic kidney disease, and a concomitant acute bacterial disease, compared to patients with mild or moderate course (OS max ≤ 4). A history of hypertension was present in 25 patients (56.8%). Among antihypertensive treatment, the use of RAAS blockers (angiotensin-converting enzyme inhibitors or angiotensin-receptor blockers) was not different between mild or moderate and severe clinical course and concerned 11 (38%) and 5 patients (33%) respectively. Beta-blockers and loop diuretics were more frequently used in patients with severe clinical progression (OS max ≥ 5) than in those with mild to moderate clinical course (OS max ≤ 4). The most common symptoms on admission were fever (86.4%) and cough (65.1%) with no differences found between patients with mild or moderate and those with severe clinical course. As expected, treatments during hospitalization including corticosteroid therapy, antibiotics and vasoactive drugs, which were significantly more frequently delivered in patients with severe compared to those with mild or moderate clinical courses. 

### 3.2. Factors Associated with Disease Severity

Laboratory findings according to disease severity during the overall period of hospitalization are described in Table 3 between mild/moderate (OS max ≤ 4) versus severe (OS max ≥ 5) groups. In univariate analyses, aldosterone levels at inclusion were significantly higher in patients with severe clinical course (OS ≥ 5) (median (min–max), 304.7 (102.5–1360.1) pmol/L) as compared to those with mild or moderate course (OS ≤ 4) (102.5 (102.5–540.2) pmol/L) (*p* < 0.01). Of note, at inclusion, potassium levels and aldosterone/renin ratios were not different between these two groups (Figure 1A,C), but in some cases, we observed a trend toward an association between higher levels of aldosterone and lower renin and potassium levels (Figure 1B,D). Concerning cortisol and ACTH levels, no difference was observed between groups. Among common hematological and inflammatory markers at baseline (including lymphocyte, monocyte and eosinophil counts, fibrinogen and D-dimers), CRP at inclusion was significantly higher in patients with severe clinical course (152 (34–389) mg/L) compared to those with mild or moderate course (83 (3–298) mg/L, *p* = 0.03). In multivariate analyses including coexisting conditions, long-term anti-hypertensive treatments, care during hospitalization and laboratory findings at inclusion, only aldosterone (OR = 1.07 (1.01–1.14), *p* = 0.033) and CRP (OR = 1.11 (1.01–1.22), *p* = 0.024) remained positively associated with the severity of clinical course.

In addition, the plasma aldosterone and CRP levels were examined according to the clinical status at three different time points (Figure 2 and Figure 3A): (i) at inclusion (Day 0), (ii) two days later (Day 2) and (iii) at the maximum ordinal scale during the overall period of hospitalization (OS max). At inclusion, aldosterone levels were not clearly associated with a specific clinical status (*p* = 0.61) (Figure 2A). However, higher aldosterone levels at inclusion were observed in patients with OS at Day 2 or OS max ≥ 5 (*p* = 0.006 and *p* = 0.0013 respectively) (Figure 2A). Moreover, aldosterone levels were also gradually and significantly increased when we compared clinical status of patients in the three following categories: mild (OS ≤ 3), moderate (OS = 4) and severe (OS ≥ 5) on Day 2 and at OS max (analysis of variance, *p* = 0.001 and *p* = 0.006, respectively) (Figure 2B). Notably, similar findings were observed when patients receiving a RAAS blocker were excluded from the analysis on Day 2 and at OS max (analysis of variance, *p* = 0.008 and *p* = 0.025, respectively) (Appendix A). Similarly, these findings were also observed when patients receiving beta-blockers were excluded from the analysis on Day 2 and at OS max (analysis of variance, *p* = 0.002 and *p* = 0.01, respectively). Notably, we also investigated the effects of age and sex on our findings and did not observe any significant differences between groups.

An additional analysis with CRP level at baseline found concordant results. As compared to patients with mild or moderate clinical status, CRP levels were significantly higher in patients with OS ≥ 5 on Day 2 or at OS max (*p* = 0.01 and *p* = 0.02, respectively) (Figure 3A). Considering the hypothesis that aldosterone may be involved in inflammatory damages of Covid-19, we searched for a relationship between the aldosterone and CRP levels. We conducted this further analysis independently from disease severity (as assessed by the OS), after having excluded the patients who had developed a documented acute bacterial infection in the days close to the biological investigation. Finally, we observed a correlation between aldosterone and CRP levels among patients with an aldosterone level greater than 102.5 pmol/L. In this group of 19 patients, aldosterone level was positively correlated with CRP level at baseline (Spearman coefficient *r* (95% CI) 0.61 (0.2–0.84), *p* = 0.006) (Figure 3B).

## 4. Discussion

In the present study based upon data collected in the real-life settings of the brutal SARS-CoV-2 outbreak, we report an association between the plasma levels of aldosterone close to admission and the severity of Covid-19 course, as defined by the ordinal scale grade. Indeed, the most severe patients, who required at least intensive care (OS ≥ 5), had significantly higher plasma levels of aldosterone when admitted than those hospitalized in medical units, with (OS = 4) or without (OS = 3) oxygen support. This association appears to be relevant both when considering the OS 2 days after admission and according to the maximal OS during the overall period of hospitalization. In most patients, aldosterone levels remained within a physiological range, but the significant differences observed between groups according to severity were independent of the renin levels and aldosterone/renin ratio. Such a hormonal profile is suggestive of a renin-independent hyperaldosteronism [10], which could be a hallmark of some patients with the most severe forms of Covid-19. Conversely, low aldosterone levels were observed in those with a less severe disease (OS = 3 or 4). This could be related either to a failure of the aldosterone assay to discriminate within the minimal values or reflect a tendency to adrenal insufficiency. However, this latter hypothesis is not supported by the plasma cortisol levels.

As previously reported [3,4,5], CRP levels were coherent with the severity of Covid-19, which is characterized by a severe inflammatory syndrome. Interestingly, patients with aldosterone levels higher than 102.5 pmol/L exhibited a linear relationship between CRP and aldosterone levels. This further finding is in line with the suspected role of the viral load in the ACE/ACE2 imbalance, which occurs before the onset of the cytokine storm [2,11,12]. Indeed, SARS-CoV-2 could disrupt the RAAS through its binding to ACE2, which is the negative regulator of the system [8]. Hence, the defective inactivation of Ang2 could lead to the activation of RAAS, including an increased secretion of aldosterone.

The role of Ang2 in the severity of lung inflammatory damage in Covid-19 is supported by previous investigational reports. First, Imai et al. [13,14] demonstrated in several animal models of acute lung injury (acid inhalation, sepsis or pneumonia) that Ang2 can worsen pulmonary lesions (including inflammatory infiltrates) through the stimulation of the Ang2 type 1 receptor (AT1R). Conversely, ACE2 and Ang2 type 2 receptor (AT2R) can down-regulate these deleterious effects, whereas abrogated ACE2 expression can induce severe respiratory failure in mice models. In addition, the levels of Ang2 are increased in these mice, which exhibit severe lung involvement partially reversible with the pharmacological inhibition of the AT1R [13,14]. During SARS-COV-1 infection, ACE2 knockout mice were resistant to virus infection and their lung samples were devoid of inflammation [15]. In contrast, the binding of the SARS-spike protein to ACE2 downregulates this regulator pathway, leading to severe lung injury and acute respiratory failure, as illustrated in a mouse model by Kuba and Coll [15]. In their study, blocking the RAAS limited the lung injury. These findings are in line with the concept that RAAS disruption could trigger inflammation in Covid-19.

Furthermore, beside coronavirus infections, the potentially deleterious effects of RAAS have been documented in several tissues (including heart and lung) and medical conditions (such as hypertension, heart failure, obesity, etc.) [16] and have been also documented beyond the regulation of sodium, extracellular volume and blood pressure. The mechanisms leading to RAAS toxicity also include (i) modulation of the production of pro-inflammatory cytokines (such as TNF alpha and IL-1 by Ang2 [17] and IL-6 by aldosterone [18]), leading potentially to recruitment of mono/macrophages; (ii) induction of fibrosis (through AT1R) [19]; and (iii) induction of vascular toxicity [20] and modulation of angiogenesis [21,22]. In the context of ACE2 neutralization by SARS-CoV-2, all these mechanisms could be exacerbated, while their clinical consequences are more limited in classical conditions of RAAS hyperactivation (such as chronic heart failure, etc. [16]). Importantly, the pathogenic mechanisms of Covid-19 are concordant with autoptic observations and biological findings, which include the cytokine storm (with IL6, IL1, TNF, etc.) [2,11,12], fibrosis [23], endothelitis and modulated angiogenesis [24,25].

In addition, the key role of RAAS toxicity could be also corroborated by the promising beneficial effects observed with anti-aldosterone and RAAS blocker treatments in several experimental conditions of pulmonary diseases [26]. In Covid-19, these protective effects are extensively debated [8,27,28]. Finally, the potentially deleterious effects of RAAS may take place in the pathophysiology of Covid-19. From this point of view, our findings suggest that both CRP and aldosterone levels may impact the clinical status. Further studies are required to document and confirm the suspected role of RAAS in Covid-19.

Our study has limitations due to the collection of plasma samples for hormonal assays in an emergency setting related to the admission for Covid-19 acute infection. Hence, optimal standardized conditions for assessing plasma renin and aldosterone levels were not met, and multiple confounding factors could be involved in the modulation of plasma aldosterone secretion. However, when we adjusted for all of these confounding parameters, plasma aldosterone levels remained significantly associated to disease severity.

In the present study, higher plasmatic aldosterone and CRP levels at inclusion are associated with severe clinical course of Covid-19 in hospitalized patients, and both parameters appear to be correlated. Our results suggest that aldosterone levels may reflect the severity of Covid-19, but this remains to be demonstrated at a larger scale. Our findings open new perspectives into the understanding of the contribution of RAAS in Covid-19 and its possible role in the outcomes of Covid-19. Further investigations are awaited to explore more thoroughly the association between increased aldosterone levels, ACE/ACE2 imbalance, inflammatory biomarkers and the severity of the Covid-19 course.

## Figures and Tables

**Figure 1 jcm-09-02315-f001:**
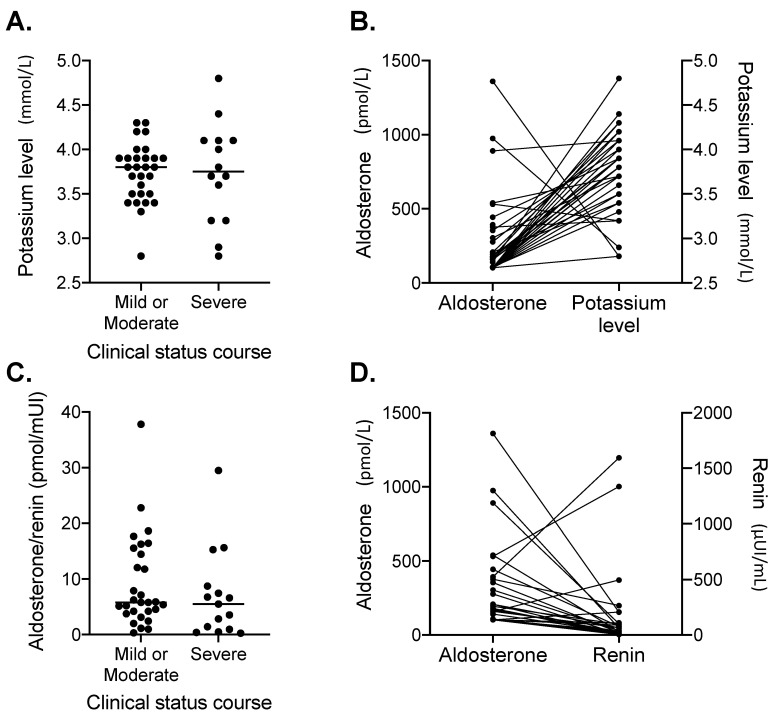
Laboratory findings of RAAS explorations. (**A**) Plasmatic aldosterone levels according to the maximal ordinal scale (OS) during the overall period of hospitalization classified in two groups: mild or moderate (OS max ≤ 4) and severe (OS max ≥ 5).; (**B**) Plasmatic aldosterone and potassium levels for each patient; (**C**) Plasmatic aldosterone / renin ratio according to the maximal ordinal scale (OS) during the overall period of hospitalization classified in two groups: mild or moderate (OS max ≤ 4) and severe (OS max ≥ 5).; (**D**) Aldosterone and renin levels for each patient.

**Figure 2 jcm-09-02315-f002:**
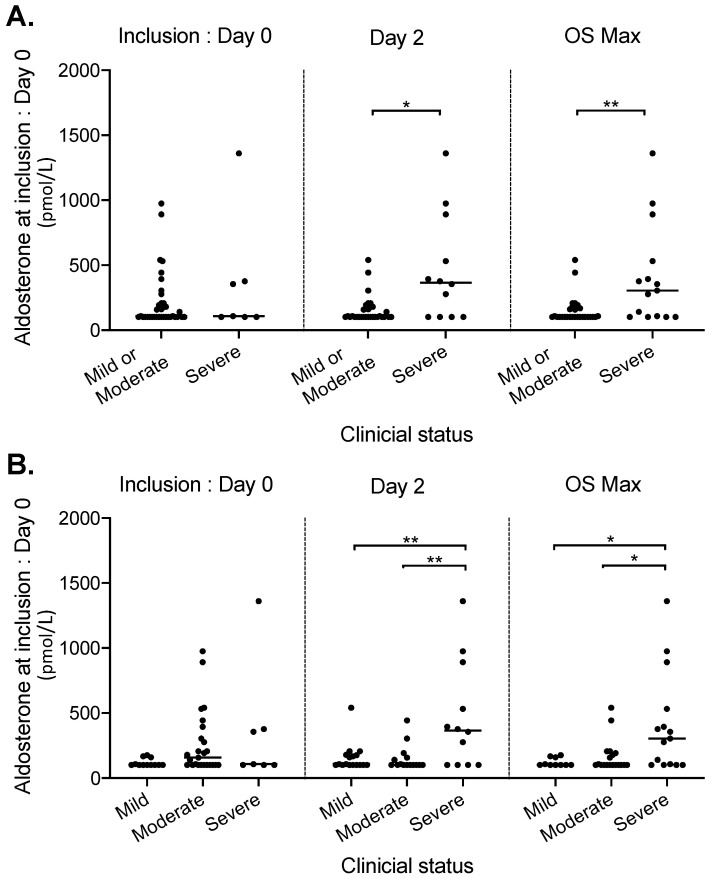
Plasma aldosterone levels at inclusion according to clinical status of patients with Covid-19 infection at Day 0, Day 2 and at the maximum ordinal scale (OS) during the overall period of hospitalization (OS max). The median values are shown (crossbar), in two (**A**) or three (**B**) groups related to clinical status severity: mild (OS ≤ 3, *n* = 10), moderate (OS = 4, *n* = 10) and severe (OS ≥ 5, *n* = 15). ** *p* < 0.01; * *p* < 0.05 (Mann–Whitney test).

**Figure 3 jcm-09-02315-f003:**
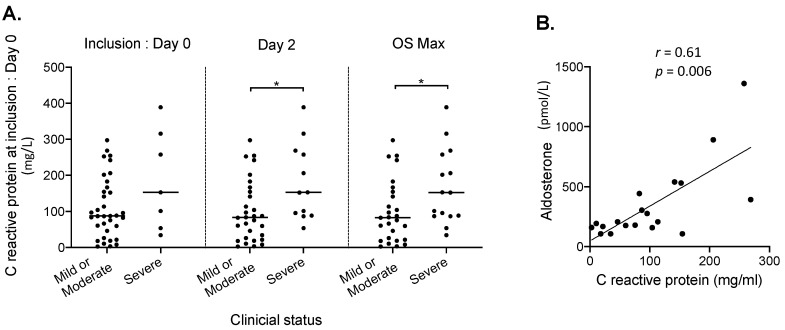
**(A**) C reactive protein level at inclusion according to clinical status of patients with Covid-19 infection at Day 0, Day 2, and maximum ordinal scale (OS) during the overall period of hospitalization (OS max). The median values are shown (crossbar), in two groups of clinical status severity: mild or moderate (OS ≤ 4, *n* = 29)) and severe (OS ≥ 5, *n* = 15). * *p* < 0.05 (Mann–Whitney test). (**B**) Correlation between aldosterone and C reactive protein at inclusion.

**Table 1 jcm-09-02315-t001:** Clinical characteristics of patients with Covid-19 according to disease severity classified in two groups: mild/moderate (OS max ≤ 4) and severe (OS max ≥ 5).

Patient Characteristics	Total (*n* = 44)	Disease Severity		*p*-Value
		Mild/Moderate (*n* = 29)	Severe (*n* = 15)	
Demographics				
Age, median (IQR)—years	66.5 (53–75.3)	65 (55–75)	70 (50.5–75.5)	0.67 ^1^
Female sex—n (%)	17 (38.6)	9 (31)	8 (53)	0.15 ^2^
**Diagnosis of Covid-19—*n* (%)**				
Positive (SARS-CoV-2 PCR)	41 (93.2)	27 (93)	14 (93)	1 ^3^
Suspected	3 (6.8)	3 (7)	1 (7)	1 ^3^
Signs and symptoms of Covid-19—*n* (%)				
Fever	38 (86.4)	27 (93.1)	11 (73.3)	0.16 ^3^
Myalgia	14 (32.6)	9 (31)	5 (35.7)	1 ^3^
Cough	28 (65.1)	20 (69)	8 (53.3)	0.31 ^2^
Breathlessness	24 (54.5)	15 (51.7)	9 (60)	0.60 ^2^
Diarrhea	12 (27.3)	7 (24.1)	5 (33.3)	0.72 ^3^
Headache	2 (4.5)	1 (3.4)	1 (6.7)	1 ^3^
Anosmia	8 (18.2)	7 (24.1)	1 (6.7)	0.23 ^3^
Dysgeusia	9 (20.5)	7 (24.1)	2 (13.3)	0.7 ^3^
Delay from onset symptoms median (IQR)—day	8 (5–10)	9 (6–11)	7 (4–8)	0.07 ^1^
Coexisting conditions—*n* (%)				
Hypertension	25 (56.8)	15 (51.7)	10 (66.7)	0.34 ^2^
Diabetes mellitus	15 (34.1)	8 (27.6)	7 (46.7)	0.21 ^2^
Obesity	11 (25)	6 (20.7)	5 (33.3)	0.47 ^3^
Cardiovascular disease	16 (36.4)	10 (34.5)	6 (40)	0.72 ^2^
Stroke	5 (11.4)	2 (6.9)	3 (20)	0.32 ^3^
Chronic kidney disease (eGFR < 60 mL/min)	7 (15.9)	2 (6.9)	5 (33.3)	0.04 ^3^
Cancer	8 (18.2)	7 (24.1)	1 (6.7)	0.41 ^3^
Dysthyroid disease	5 (11.4)	1 (3.4)	4 (26.7)	0.04 ^3^
History of organ transplantation	4 (9.1)	1 (3.4)	3 (20)	0.11 ^3^
Documented acute bacterial infection	10 (22.7)	3 (10.3)	7 (46.7)	0.02 ^3^
Long-term anti-hypertensive treatment—*n* (%)				
Angiotensin-converting enzyme inhibitors	10 (22.7)	6 (20.7)	4 (26.7)	0.71 ^2^
Angiotensin-receptor blockers	6 (13.6)	5 (17.2)	1 (6.7)	0.65 ^2^
Calcium-channel blockers	6 (13.6)	5 (17.2)	1 (6.7)	0.65 ^2^
Beta-blockers	14 (31.8)	6 (20.7)	8 (53.3)	0.04 ^2^
Thiazide diuretics	3 (6.8)	2 (6.9)	1 (6.7)	1 ^2^
Loop diuretics	7 (15.9)	2 (6.9)	5 (33.3)	0.04 ^2^
Care during hospitalization				
Duration mean (SD)—day	13.1 (10.9)	10.1 (7.5)	18.8 (14.1)	0.03 ^4^
Corticosteroid therapy mean (SD)—day	1.8 (3.1)	1.1 (2.4)	3.33 (3.8)	0.02 ^4^
Use of antibiotic agents—n (%)	30 (68.2)	16 (55.2)	14 (93.3)	0.02 ^3^
Use of vasoactive drug—n (%)	7 (15.91)	0 (0.00)	7 (46.67)	<0.01 ^3^

OS denotes ordinal scale, IQR interquartile range, n number, eGFR estimated glomerular filtration rate. * Comparison mild/moderate versus severe using Student *t*-test (^1^). Chi-squared test (^2^), Fisher’s exact test (^3^) or Mann–Whitney test (^4^).

**Table 2 jcm-09-02315-t002:** Clinical course during hospitalization according to the Covid-19 ordinal scale (mild, moderate, severe gradation).

Seven-Category Ordinal Scale		Day 0 (at Inclusion)	Day 2	OS Max
Mild clinical status—*n* (%)		12 (27.3)	17 (38.6)	10 (22.7)
1	Not hospitalized, no limitations on activities—*n*	-	-	-
2	Not hospitalized, limitations on activities—*n*	-	2	-
3	Hospitalized, not requiring supplemental oxygen—*n*	12	15	10
Moderate clinical status—*n* (%)		25 (56.8)	15 (34.1)	19 (43.2)
4	Hospitalized, requiring supplemental oxygen—*n*	25	15	19
Severe clinical status—*n* (%)		7 (15.9)	12 (27.3)	15 (34.1)
5	Hospitalized, on non-invasive ventilation or high flow oxygen devices—*n*	1	5	3
6	Hospitalized, on invasive mechanical ventilation or ECMO—*n*	6	5	6
7	Death—*n*			

ECMO denotes extracorporeal membrane oxygenation and eGFR denotes estimated glomerular filtration rate using CKD-EPI (Chronic Kidney Disease - Epidemiology Collaboration) equation.

**Table 3 jcm-09-02315-t003:** Biological findings at inclusion (on Day 0) according to disease severity of patients with Covid-19 infection classified in two groups: mild/moderate (OS max ≤ 4) and severe (OS max ≥ 5).

Measurements	Number Data Available	Total	Disease Severity		Median Difference (95% CI)	*p*-Value *
			Mild/Moderate (*n* = 29)	Severe (*n* = 15)		
Univariate analysis						
C-reactive protein median (min-max)—mg/L	42	92 (3–389)	83 (3–298)	152 (34–389)	65 (6; 127)	0.03 ^1^
Lymphocyte count median (min-max)—10^9^/L	42	1.2 (0.2–2.6)	1.1 (0.2–2.6)	1.3 (0.2–2)	0.2 (−0.2; 0.5)	0.22 ^1^
Monocyte count median (min-max)—10^9^/L	42	0.5 (0.1–2.3)	0.5 (0.1–2.3)	0.5 (0.2–1.5)	0.1 (−0.1; 0.3)	0.42 ^1^
Eosinophil count median (min-max)—10^9^/L	42	0.1 (0–0.3)	0.1 (0–0.3)	0 (0–0.3)	−0.02 (−0.1; 0)	0.11 ^1^
Fibrinogen median (min-max)—g/L	39	5.7 (1.2–9.4)	5.6 (1.2–9.4)	5.9 (3.4–7.6)	0.4 (−0.8; 1.5)	0.56 ^2^
D-dimers median (min-max)—μg/L	35	870 (200–4000)	738 (372–4000)	1112 (200–4000)	215 (−247; 760)	0.32 ^1^
NT pro-BNP median (min-max)—ng/L	31	423 (17–63245)	228 (17–16749)	1135 (58–63245)	711.50 (−17; 2065)	0.06 ^1^
Troponin median (min-max)—ng/L	39	13.8 (1.5–1596)	12.4 (1.5–1596)	23.8 (1.5–24)	7.55 (−3; 68.5)	0.18 ^1^
Ferritin median (min-max)—µg/L	35	814 (133–12460)	603 (148–12460)	966 (133–7750)	217 (−330; 965)	0.43 ^1^
Potassium level median (min-max)—mmol/L	43	3.8 (2.8–4.8)	3.8 (2.8–4.3)	3.8 (2.8–4.8)	3.8 (2.8; 4.8)	0.99 ^2^
Creatinine median (min-max)—µmol/L	44	74.5 (34–949)	71 (34–949)	100 (40–524)	28 (5; 54)	0.02 ^1^
eGFR median (min-max) —mL/min/1.73 m^2^	44	87.5 (3–120)	91 (3–120)	59 (9–120)	−26 (−50; 0)	0.05 ^1^
Aldosterone median (min-max) – pmol/L	44	108 (102.5–1360.1)	102.5 (102.5–540.2)	304.7 (102.5–1360.1)	174.5 (0; 274.2)	<0.01 ^1^
Renin median (min-max) – µui/mL	44	25.8 (4.5– 1594)	19.00 (4.5–495.4)	62.4 (9.4–1594)	32.4 (9.2; 116.9)	<0.01 ^1^
Cortisol median (min-max) – nmol/mL	34	471.8 (110.9–1320.8)	495.2 (110.9–933.2)	377.5 (115.6–1320.8)	−91.8 (−259.4; 90.5)	0.17 ^1^
ACTH median (min-max)– pmol/L	36	3.3 (0.2–13.8)	5.5 (0.2–13.8)	1.7 (0.2–11.1)	−2.9 (−5.2; −0.3)	0.6 ^1^

eGFR denotes estimated glomerular filtration rate using CKD-EPI equation. * Comparison Mild/Moderate versus Severe using Mann-Whitney test (^1^) or Student *t*-test (^2^).

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
