# Peer review of "The Plasmatic Aldosterone and C-Reactive Protein Levels, and the Severity of Covid-19: The Dyhor-19 Study"

_jcm, 2020, doi:10.3390/jcm9072315_

Round 1
Reviewer 1 Report
Comments to the Author
In this manuscript, the authors evaluated plasma renin and aldosterone levels in a series of consecutively admitted patients for COVID-19 (n=44). Aldosterone and C-reactive protein (CRP), significantly higher and associated in patients with a severe clinical course as compared to mild-moderate symptom subjects. They concluded it might open new perspectives in the understanding of ACE/ACE2 imbalance and its possible role in the outcomes of COVID-19. They suggested from the study point; further investigations are awaited to explore the association between aldosterone and COVID-19 severity. The results are of interest, and the manuscript is overall well written. However, there are some suggestions and concerns, which should be addressed.
Comments
- The title may be changed to seems to be overstated, even though ‘suggests’ word used. Technically, ACE/ACE2 plasma soluble forms not measured and beyond CRP, nothing measured to show inflammatory injury in this study. It is interesting to see the plasma ACE2, Ang2, and angiotensin peptide. The title may be has rewritten more precisely, and “maybe CRP included.”
- Abstract: Conclusions part is different from the study outcome and discussion portion. Conclude the critical outcome of the study and maybe a few lines of perspectives and suggestions included at the end of the conclusion (both in abstract and final manuscript conclusions portions must be modified).
- Introduction: Include a few lines about CRP.
- Study design: Dyhor-19 Line 80: hormon’e’ Is that letter ‘e’ missing?
- Data collection: Kit’s catalog number, inter-assay, and intra-assay co-efficient those details must be provided.
- Results: From table1: Half of the subjects are hypertensive; how anti-hypertensive treatment does not affect the studied parameter (Lines145-150)? Those drugs affect the RAAS. Why not affected, did authors analyzed those subjects separately? What kind of analysis used to rule out anti-hypertensive treatment?
- In what way is CRP superior instead of other markers?
- Figure3B: r value must be mentioned.
- Does authors observed any gender bias in this study outcomes; between, male, and female. Since COVID-19 affects more males than females?
- Line262: “In the present study, our data…… potential deleterious…...” This is an overstatement for the observed parameters. It may be rewritten more precisely.
- There are many other minor errors of syntax and grammar throughout the text, which need to be fixed. Abbreviations must be defined in the first instance. Author contributions and funding statements missing.
Author Response
We thank the reviewer for these comments and questions.
- The title may be changed to seems to be overstated, even though ‘suggests’ word used. Technically, ACE/ACE2 plasma soluble forms not measured and beyond CRP, nothing measured to show inflammatory injury in this study. It is interesting to see the plasma ACE2, Ang2, and angiotensin peptide. The title may be has rewritten more precisely, and “maybe CRP included.”
We agree with the reviewer that the title may be considered as overstated. So, we modified the title and included the term CRP. Our proposal is now : “The plasmatic aldosterone and C-Reactive Protein levels and the severity of COVID-19 (Dyhor-19 Study)”. We hope that this title does correspond better to the content of our study and will fit the reviewer’s request.
2. Abstract: Conclusions part is different from the study outcome and discussion portion. Conclude the critical outcome of the study and maybe a few lines of perspectives and suggestions included at the end of the conclusion (both in abstract and final manuscript conclusions portions must be modified).
We fully agree with the reviewer’s comment. Consequently, we modified the conclusion both in abstract and manuscript, as follows:
- Abstract: “ Both plasmatic aldosterone and CRP levels at inclusion are associated with the clinical course of Covid-19. Our findings may open new perspectives in the understanding of the possible role of RAAS for Covid-19 outcome. “
- Manuscript lines 364-371: “ In the present study, higher plasmatic aldosterone and CRP levels at inclusion are associated with severe clinical course of Covid-19 in hospitalized patients, and both parameters appear to be correlated. Our results suggest that aldosterone levels may reflect the severity of Covid-19, but this remains to be demonstrated at a larger scale. Our findings open new perspectives in the understanding of the contribution of RAAS in Covid-19 and its possible role in the outcomes of Covid-19. Further investigations are awaited to explore more thoroughly the association between increased aldosterone levels, ACE/ACE2 imbalance, inflammatory biomarkers and the severity of Covid-19 course.
3. Introduction: Include a few lines about CRP.
We modified the introduction section and included a new sentence on CRP :
Lines 67-69 : “In this context, the intensity of the inflammatory process contributes to the disease severity and the plasmatic level of C-Reactive protein (CRP) (a biomarker of systemic inflammation) could represent a marker of bad outcome in Covid-19 patients [3–6]. “
We propose to include herein the references 7, 8 and 9 from the old version. We also propose to add a new reference by Cecconi et al, recently published in the JCM.
4. Study design: Dyhor-19 Line 80: hormon’e’ Is that letter ‘e’ missing?
We agree with the reviewer and modified the manuscript accordingly (Line 86).
5. Data collection: Kit’s catalog number, inter-assay, and intra-assay co-efficient those details must be provided.
According to the reviewer’s comment, we modified the manuscript and included the required information as follows:
Lines 119-125 : “Determination of CRP was performed on the Cobas8000/e502® analyzer (Roche Diagnostic, Meylan, France) using immunoturbidimetric method with reagents from ROCHE (total CV imprecision results in the laboratory = 3%). Renin and aldosterone were determined on an IDS-iSYS multi-discipline automated system (ImmunoDiagnosticSystem, Boldon, United Kingdom) using kits from IDS (total CV imprecision results in the laboratory 5% and 6% for Renin and aldosterone respectively).”
6. Results: From table1: Half of the subjects are hypertensive; how anti-hypertensive treatment does not affect the studied parameter (Lines145-150)? Those drugs affect the RAAS. Why not affected, did authors analyzed those subjects separately? What kind of analysis used to rule out anti-hypertensive treatment?
We thank the reviewer for this crucial comment. As indicated in lines 210-218 and supplemental figure 1, we also performed additional sets of analysis in which patients receiving RAAS blockers and beta blockers respectively were excluded. Similar findings were observed when these patients were excluded from the analysis. We have also generated a figure concerning the exclusion of patients receiving Beta blockers (Supplemental figure 2, please see attached files). We did not plan to publish it, except if this is the reviewer's wish.
7. In what way is CRP superior instead of other markers?
Several other markers (including lymphocyte, monocyte and eosinophil counts, fibrinogen and D-dimers) were investigated in our patients. As indicated in lines 194-201, CRP was the sole ‘usual’ biomarker to discriminate between groups (in univariate and multivariate analysis).
8. Figure3B: r value must be mentioned.
We modified this figure accordingly.
9. Does authors observed any gender bias in this study outcomes; between, male, and female. Since COVID-19 affects more males than females?
We thank the reviewer for this important comment. Our manuscript was unclear on this point. Indeed, we investigated the effects of age and sex on our findings and did not observe any differences between groups (age: p-value = 0.6957 and sex: p-value = 0.2656). So, we modified the manuscript as follows: - patients and methods section(Lines 138-39) : “ Potential confounding factors were investigated by testing differences between groups”.
- Results section (Lines 218-219): we investigated the effects of age and sex on our findings and did not observe any differences between groups.
10. Line262: “In the present study, our data...... potential deleterious......” This is an overstatement for the observed parameters. It may be rewritten more precisely.
We also thank the reviewer for this comment and agree that our sentence was particularly unclear and unprecise. So, we modified the paragraph as follows:
Lines 307-310: “In addition, the key role of RAAS toxicity could be also corroborated by the promising beneficial effects observed with anti-aldosterone and RAAS blockers treatments in several experimental conditions of pulmonary diseases [25]. In Covid-19, these protective effects are extensively debated [4,26,27]. Finally, the potentially deleterious effects of RAAS may take place in the pathophysiology of Covid-19. From this point of view, our findings suggest that both CRP and aldosterone levels may impact the clinical status. Further studies are required to document and confirm the suspected role of RAAS in Covid-19.
11. There are many other minor errors of syntax and grammar throughout the text, which need to be fixed. Abbreviations must be defined in the first instance. Author contributions and funding statements missing.
We revised the manuscript according to the reviewer’s comments and hope it will now fit with the requests.
- Study funding: This research received no external funding
- Author contributions:
Orianne VILLARD : data collection, design and conceptualization of the study, statistical analysis, drafting and reviewing of the manuscript ; David MORQUIN : data collection, investigation, software, design and conceptualization of the study, statistical analysis, supervision, drafting and reviewing of the manuscript; Nicolas MOLINARI : methodology, statistical analysis, drafting and reviewing of the manuscript; Isabelle RAINGEARD : design and conceptualization of the study, drafting and reviewing of the manuscript; Jean-Paul CRISTOL : data collection, drafting and reviewing of the manuscript ; Boris JUNG : data collection, investigation, drafting and reviewing of the manuscript; Camille ROUBILLE : data collection, investigation, drafting and reviewing of the manuscript ; Vincent FOULONGNE : data collection, investigation, drafting and reviewing of the manuscript ; Pierre FESLER : data collection, investigation, drafting and reviewing of the manuscript; Patrice TAOUREL : data collection, investigation, drafting and reviewing of the manuscript ; Amadou KONATE : data collection, investigation, drafting and reviewing of the manuscript ; Alexandre Thibault Jacques MARIA : design and conceptualization of the study, drafting and reviewing of the manuscript ; Alain MAKINSON : data collection, investigation, drafting and reviewing of the manuscript ; Ivan BERTCHANSKY : data collection, investigation, drafting and reviewing of the manuscript ; Romaric LARCHER : data collection, investigation, drafting and reviewing of the manuscript; Kada KLOUCHE : data collection, investigation, drafting and reviewing of the manuscript; Vincent LE MOING : data collection, investigation, drafting and reviewing of the manuscript; Eric RENARD : design and conceptualization of the study, methodology, drafting and reviewing of the manuscript; Philippe GUILPAIN: design and conceptualization of the study, methodology, statistical analysis, supervision, drafting and reviewing of the manuscript

Reviewer 2 Report
Investigations in the Renin–Angiotensin–Aldosterone System in COVID-19 are critical for better understanding of the disease and improving patient care. To date direct measurements of angiotensin 2 in COVID patients have rarely been reported. In this study, Villard et al. reported the association of the level of plasma aldosterone early after admission with the clinical course of the disease. Although the patient cohort is limited and the findings need to be verified in larger studies, this study provides new evidence for the involvement of RAAS in the disease development of COVID and the potential modulation of this system for the clinical care.
Revisions:
- On line 167-170, apparently gender and age are not included in the multivariate analysis. Because the sample size is limited, these potential confounding factors should be investigated, although they are not significant individually between the patient groups.
- Patient information can be better presented in Table 1, perhaps by organizing the information into groups, e.g. demographics, comorbidity, etc.
Author Response
We thank the reviewer for these comments.
Revisions:
- On line 167-170, apparently gender and age are not included in the multivariate analysis. Because the sample size is limited, these potential confounding factors should be investigated, although they are not significant individually between the patient groups.
We thank the reviewer for this important comment. Our manuscript was unclear on this point. Indeed, we investigated the effects of age and sex on our findings and did not observe any differences between groups (age: p-value = 0.6957 and sex: p-value = 0.2656). So, we modified the manuscript as follows: - patients and methods section (Lines 138-39) : “ Potential confounding factors were investigated by testing differences between groups”.
- Results section (Lines 218-219): we investigated the effects of age and sex on our findings and did not observe any differences between groups.
- Patient information can be better presented in Table 1, perhaps by organizing the information into groups, e.g. demographics, comorbidity, etc.
We modified Table 1 accordingly and hope that it is clearer (please see manuscript or attached file)
